# Applying Speckle Noise Suppression to Refractive Indices Change Detection in Porous Silicon Microarrays

**DOI:** 10.3390/s19132975

**Published:** 2019-07-05

**Authors:** Ruyong Ren, Zhenhong Jia, Jie Yang, Nikola Kasabov

**Affiliations:** 1College of Information Science and Engineering, Xinjiang University, Urumuqi 830046, China; 2Institute of Image Processing and Pattern Recognition, Shanghai Jiao Tong University, Shanghai 200240, China; 3Knowledge Engineering and Discovery Research Institute, Auckland University of Technology, Auckland 1020, New Zealand

**Keywords:** PSi, PNLM, median filtering, speckle noise

## Abstract

The gray value method can be used to detect gray value changes of each unit almost parallel to the surface image of PSi (porous silicon) microarrays and indirectly measure the refractive index changes of each unit. However, the speckles of different noise intensities produced by lasers on a porous silicon surface have different effects on the gray value of the measured image. This results in inaccurate results of refractive index changes obtained from the change in gray value. Therefore, it is very important to reduce the influence of speckle noise on measurement results. In this paper, a new algorithm based on the concepts of probability-based nonlocal-means filtering (PNLM), gradient operator, and median filtering is proposed for gray value restoration of porous silicon microarray images. A good linear relationship between gray value change and refractive index change is obtained, which can reduce the influence of speckle noise on the gray value of the PSi microarray image, improving detection accuracy. This means the method based on gray value change detection can be applied to the biological detection of PSi microarray arrays.

## 1. Introduction 

The biochip (microarray chip) is a new technology developed by research of the human genome project. Because of their excellent characteristics, extensive application prospects, and rapid development, biochips have demonstrated great application value in disease diagnosis, drug development, genetic modification, allergen detection, and environmental protection since they were first proposed in the 1990s [1]. Biochips are an effective means for people to obtain biological information efficiently and on a large scale. So far, commercial bio-microarray chip detection is still based on fluorescence and isotope-labeling methods. However, fluorescent labeling requires sophisticated reagent selection and pairing, in addition to reagent modification including synthesis and purification, which potentially changes the intrinsic properties of the capture probe and/or target molecules [2] and dramatically increases the cost and complexity of the assays. Therefore, non-labeled detection technology has gradually become an important development direction in the field of biological analysis in recent years [3,4,5]. Among many label-free detection technologies, the image gray value detection method of porous silicon photonic crystal bio-microarray is a new parallel quantitative detection technology of bio-array chips that is label-free, has high measurement sensitivity, and is cost-effective [6,7]. 

Porous silicon is an excellent biomaterial with a large specific surface area, a large interaction area between biological molecules, and high sensitivity for sensing and detection. Porous silicon has very good bioactivity, compatibility, and adsorption, allowing biological molecules to enter the device [8]. Porous silicon photonic devices are used to measure the increase in the refractive index of the device caused by the reaction between the target biological molecules and the probe molecules fixed in the inner wall of the holes, which can detect the target organism quantitatively without labeling. Among all kinds of porous silicon biosensors with photonic structure, the PSi microcavity biosensor based on photonic crystal resonance structure has the high sensitivity and has been used in various kinds of biological reaction detection [9,10,11]. As the reflection or transmission spectrum of the microcavity is very sensitive to the incident angle of incident light, a biological detection method without using a spectrometer was developed, which detects the refractive index change caused by a biological reaction by measuring the change of reflected or transmitted light intensity with the incident angle. The PSi microcavity demonstrates high-sensitivity biological detection based on the reflection angle spectrum and the transmission angle spectrum [12,13]. No spectrophotometer is needed in the detection process; only one laser and an angle measuring instrument are needed in the biological detection. The change of the refractive index can be measured by the gray value changes of reflection or transmission images of the PSi microcavity acquired by CCD (Charge-coupled Device) the on angle spectrum method [14,15]. Although the sensitivity of the detection has somewhat decreased, it can be applied to the parallel detection of the PSi microcavity microarrays. 

In order to detect the image gray value easily, the array image is preprocessed, tilt-corrected, and segmented. After such processing, the gray value of each array unit can be calculated automatically by a computer [16]. 

Due to the rough surface of PSi, interference among the scattered laser rays results in speckle noise on CCD [17]. This noise not only reduces the resolution and contrast of the image, but also causes changes in the image gray values [18]. Before the biological reaction, each unit of the PSi array is modified and probe biomolecules (such as DNA or antibodies) are immobilized in the PSi. After the reaction, the different reactant in each unit formed by the specific binding of different target biological molecules (such as complementary DNA or antigens) with the probe molecules are fixed in the PSi. These different kinds of reactants with different concentrations will result in different morphologies of porous silicon surface, thus, the speckle noise intensity in each unit is different. These different speckle noise intensities will cause different amounts of interference with the gray values of the images. This results in inaccurate refractive index changes based on gray value changes. To solve this problem, the influence of speckle noise on the image gray values is analyzed and a method of reducing the influence of speckle noise on the gray values of PSi microarray images is proposed. The experimental results show that the proposed method can restore the gray value affected by speckle noise very well.

## 2. Theoretical Analysis

### 2.1. Speckle Noise Model

In PSi microarray images, speckle noise is caused by the interaction between a laser light and the roughness of the PSi surface. The existence of speckle noise is inherent to this detection method and it exhibits distinctive characteristics in terms of its certainty and randomness. Speckle noise can be modeled as multiplicative noise, with the obtained signal being a product of the original signal and the speckle noise [19]. Let *I*(*i*, *j*) denote a distorted pixel in an image and let *M*(*i*, *j*) denote the corresponding noiseless image pixel. According to the multiplicative noise model,
(1)I(i,j)=M(i,j)×N(i,j),
where *N*(*i*, *j*) represents the speckle noise signal [19].

Speckle noise also affects the gray values of the image itself. The larger the variance of the speckle noise is, the greater its influence on the gray values of the image and the more difficult it is to recover the noiseless image. To better explain the influence of speckle noise, the effects of speckle noise with variances of 0.1 and 0.9 on an image with a gray value of 40 are shown in Figure 1.

If the mean values of the *N* normal speckles participating in the superposition are denoted by *I_k_*, *K* = 1, 2, …, *N*, then the probability density function of the intensity of the *k*th normal speckle, *I_k_*, is an exponential distribution [20], as shown in Formula (2):(2)$$P_{I_{k}}(I_{k})=\left\{\begin{array}{ll} \frac{1}{\bar{I_{k}}}\exp(-\frac{I_{k}}{I_{k}}) & I_{k}\geq 0\\ 0\; & I_{k}<0 \end{array}\right..$$

Because the *I_k_* are statistically independent, the probability density function of their sum can be calculated as *N*-1 reconvolutions of the corresponding function *P_I_*(*I*), as shown in Formula (3):(3)$$P_{I}(I)=P_{I_{1}}(I)⊗P_{I2}(I)⊗\cdot \cdot \cdot ⊗P_{I_{k}}(I).$$

If the *N*, *I_k_* are not zero and are not equal to each other, then Formula (2) can be rewritten as follows:(4)$$P_{I}(I)=\left\{\begin{array}{ll} \frac{I^{\left(N-1\right)}}{\Gamma (N)I_{0}{}^{N}}\exp(-\frac{I}{I_{0}}) & I\geq 0\\ 0\; & I<0 \end{array}\right..$$

The probability density functions of *N* different independent speckle patterns of equal intensity are shown in Figure 2. As *N* changes with the average total strength remaining unchanged, the probability density function changes from a negative exponential distribution (N = 1) to a Gaussian density function distribution (N = 10), which is consistent with the central limit theorem. 

### 2.2. Influence of Speckle Noise 

Figure 3a shows an original image with a gray value of 40 at each pixel. Speckle noise of different intensities (variances) was added to this original image in MATLAB (2016a, MathWorks, Natick, MA, USA) to generate noisy experimental images. The variance (Var) of the noise was varied between 0 and 1 in intervals of 0.1; thus, 10 images with different speckle intensities were generated. The corresponding variances are 0.1, 0.2, 0.3, 0.4, 0.5, 0.6, 0.7, 0.8, 0.9, and 1.0. In Figure 3b–f, five of the resulting noisy images are presented. 

In research on speckle noise, previous works have evaluated denoising algorithms in two ways [21,22,23]. Subjective evaluations concern how the textural details and visual effect of an image are improved after denoising. Objective evaluations rely on quantitative metrics, such as the speckle index (SI), peak signal-to-noise ratio (PSNR), equivalent number of looks (ENL), and structural similarity index (SSIM).

For the denoising of PSi microarray images, however, the subjective effects and traditional objective indicators are not important. Because the final detection result depends only on the average gray value of each cell in the array, these gray values need to be restored to remove the influence of the noise to enable accurate detection. At present, there are no related works on the restoration of gray values affected by speckle noise. To study the effects of speckle noise on the image gray values, we added speckle noise with variances of 0–1 to 2500 images with a gray value of 40 and obtained the corresponding relationship between the resulting average gray value and the speckle intensity, as shown in Figure 4c. As the speckle intensity increased, the average gray value of the image increased. The same result was found after testing three other sets of 2500 images each, with initial gray values of 20, 30, and 50. 

To explain the reason for the observed relationship between the speckle intensity and the average gray value, in this paper, 7500 speckle noise images were tested and the relationship between speckle intensity and singular pixel points (bright spots with an abnormal increase in gray value) of the images was drawn, as shown in Figure 5. Each of the curves in Figure 5 represents the relationship between speckle intensity and singular pixel points of an image. By comparing Figure 4 and Figure 5, it can be seen that the variation in the average gray value of the image is related to the number of singular pixels. When the speckle intensity was within the range of 0–0.3, the number of bright spots was very small. Hence, the randomness of the scattered noise became the main factor affecting the average gray value of the image, causing the average gray value to fluctuate around the original gray value. As the speckle intensity increased beyond 0.3, the number of bright spots gradually increased and these bright spots became the main factor affecting the average gray value of the image, resulting in an overall increase in the average gray value. 

The above conclusions are very important for the detection of biological reactions in PSi microarrays based on digital images. In this image-based detection method, the changes in the refractive index that were caused by biological reactions were determined by measuring the changes in the gray values of the PSi microarray images. Speckle noise produced by laser irradiation will always be present in PSi microarray images obtained directly with digital imaging equipment, but the speckle noise intensities will be different for PSi surface images obtained before and after a reaction. According to the findings reported above, the higher the speckle noise intensity was, the greater its impact on the image gray values. Thus, the changes in the gray values measured in the biological detection experiment were partly due to the true changes in the refractive index of the PSi before and after the reactions and partly due to image speckle noise. Therefore, the ability to remove speckle noise in PSi microarray images is very important for the detection of biological reactions. 

## 3. Theoretical Analysis

### 3.1. Algorithm Description

Many excellent algorithms have emerged from research on speckle noise. The classical algorithms include common Lee filters, Kuan filters, mean filters, and median filters [24,25,26]. Previous works have introduced some improvements to these classical algorithms and have achieved good results [27,28,29,30,31,32]. Recently, scholars have also proposed advanced algorithms, such as probability-based nonlocal means filtering [33], numerical multilook and 3D block-matching filtering [34], adaptive wavelet threshold processing [35], and adaptive anisotropic diffusion [36]. In this paper, a variety of filtering algorithms were experimentally analyzed. Most algorithms show good performance with respect to common evaluation indexes, such as the SI, PSNR, and ENL, but they cannot modify the gray value of an image. Therefore, these algorithms cannot restore the gray value in the presence of speckle noise. The weights in the BM3D and probability-based nonlocal-means filtering (PNLM) algorithms are determined by the distances between similar blocks and the gray value of a target pixel is obtained by weighting the gray values of all the pixels in its neighborhood. A large number of scattered spots and speckle blocks increases the gray value differences between two neighborhood windows and thus affects the accuracy of the weights. The gradients of images with scattered spots and speckle blocks are large and the good edge retention of the PM algorithm results in incorrectly identified edges and a poor denoising effect. Large numbers of scattered spots and speckle blocks similarly affect the selection of the optimal threshold for adaptive wavelet transform, thereby worsening the denoising effect. For the classical median and adaptive median algorithms, large numbers of scattered spots and speckle blocks result in a large difference between the selected median pixel and the original pixel, affecting the denoising process. 

Many classical NLM algorithms and some improved NLM algorithms have good effects on the removal of additive Gaussian noise, but they are not suitable for multiplicative speckle noise. In order to solve this problem, Hancheng Yu et al. proposed a probability-based NLM algorithm [33]. NLM filtering algorithm uses a larger search window or a weighted average of the pixels in the whole image to replace the pixels considered, as shown in Formula (5):(5)$$\overset{\wedge }{u}=\frac{\sum_{j\in B(i,r)}w_{i,j}u_{j}}{\sum_{j\in B(i,r)}w_{i,j}},$$
where u represents the restoration value at the pixel point i and B (i, r) represents the search window with the center of i dimension and the size of (2r + 1) × (2r + 1) pixels. Since NLM filters use non-local self-similarity to preserve the characteristics of the pixels, weights w_i,j_ are used to measure the similarity between the central pixels of the search window. The values of weights w_i,j_ depend on the square Euclidean distance between two pixels. Each pixel value is restored to the average value of the most similar pixel. Considering that the characteristics of these damaged pixels are similar to impulse noise, the method of identifying impulse noise pixels can be used to suppress speckle noise. Define an absolute deviation of pixel strength d_i,j_, as shown in Formula (6).
(6)$$d_{i,j}=|u_{i}-u_{j}|\;j\in B(i,f),$$
where u_i_ and u_j_ are the intensity of pixels i and j respectively. Garnett et al. introduced a local image sorting absolute difference (ROAD) [37] for statistical recognition of impulse noise pixels. The definition of ROAD is shown in Formula (7).
(7)$${\mathrm{ROAD}}_{L}(u_{i})=\sum_{m=1}^{L}r_{m},$$
where r_m_ represents the mth minimum value of d_i,j_. ROAD statistics provide the proximity of a pixel value to its nearest neighbor. The basic idea of this statistic is that the intensity of unnecessary pulses is very different from most or all of their adjacent pixels. For impulsive noise pixels, statistics are used to calculate the probability, Pi, of each pixel not damaged, as in Formula (8).

(8)$$P_{i}=\exp(-0.01{\mathrm{ROAD}}_{L}(u_{i}))$$

The calculated probability shows the distance between two pixel blocks. The corresponding pixel points in each pixel block have a high undamaged probability, which is used to calculate a larger weight in the process of calculating the distance between pixel blocks. The distance between pixel blocks based on probability is shown by Formula (9).
(9)$$d_{i,j}^{2}(B(i,j),B(j,f))=\frac{1}{{\left(2f+1\right)}^{2}}\sum_{k\in B(0,f)}{\left(p_{i+k,j+k}(u_{i+k}-u_{j+k})\right)}^{2},$$
where p_i+k_, _j+k_ denotes the minimum value between p_i+k_ and p_j+k_.

Therefore, the calculated non-damage probability can reduce the impact of pixel damage and can more accurately calculate the similarity between pixel blocks. Then, the probability is used to restore the central pixels, as shown in Formula (10).

(10)$$\overset{\wedge }{u_{i}}=\frac{\sum_{j\in B(i,x)}w_{i,j}p_{j}u_{j}}{\sum_{j\in B(i,x)}w_{i,j}p_{j}}$$

In the process of recovery, for each pixel in the search window, the weighted average based on probability will eliminate the severely damaged data. 

The existence of small gray deviation affects the good linear relationship between gray value change and refractive index change and affects the measurement of refractive index in the experiment. So, it is very important to restore the correct gray value of speckle noise image in the experiment. The smaller the gray value error of the recovered speckle image, the more accurate the refractive index obtained in the experiment. However, speckle noise is complex and stubborn; it is difficult to suppress speckle noise and restore the gray value of the image. Even for simple texture images such as porous silicon microarray, the process of restoring their accurate gray values is difficult. The residual speckle noise after PNLM treatment cannot be eliminated and it is not feasible to use simple PNLM. Since PNLM removes most of the noise, the remaining speckle noise can be further removed with other algorithms to complete the gray value restoration of speckle noise. Therefore, this paper improves the PNLM algorithm by introducing the concept of a gradient operator into the PNLM algorithm. According to the change of the image gradient, residual speckle noise points were marked. The marked speckle was used as the center of the median filter window and the new center value of the median filter was used to replace the original center value. Repairing only the marked pixels saves a lot of time and achieves efficient and fast gray value recovery. For convenience, we named the algorithm proposed in this paper as PNLM-G. 

The gradient magnitudes and directions for 8 neighborhoods in the smoothed gray image were calculated using a 9 × 9 template. Thresholds were applied to identify not only the speckle points but also the edges of speckle blocks based on the first-order partial derivatives at each pixel, which are calculated as follows. 

Partial derivative in the X direction:(11)$$G_{x}(x,y)=S(x+1,y)-S(x-1,y).$$

Partial derivatives in the Y direction:(12)$$G_{y}(x,y)=S(x,y+1)-S(x,y-1).$$

Partial derivatives in the 45° direction: (13)$$G_{45^{{}^{\circ}}}(x,y)=S(x-1,y+1)-S(x+1,y-1).$$

Partial derivatives in the 135° direction: (14)$$G_{135^{{}^{\circ}}}(x,y)=S(x+1,y+1)-S(x-1,y-1).$$

The expression of gradient amplitude calculated by second-order Euclidean norm is as follows:(15)$$G(x,y)=\sqrt{G_{x}{\left(x,y\right)}^{2}+G_{y}{\left(x,y\right)}^{2}+G_{45^{{}^{\circ}}}{\left(x,y\right)}^{2}+G_{135^{{}^{\circ}}}{\left(x,y\right)}^{2}}.$$

The calculation expression of gradient direction is as follows:(16)$$\theta (x,y)=\tan^{-1}(G_{y}(x,y)/G_{x}(x,y)),$$
where *G_x_*(*x*, *y*) is the gradient at point (x, y) in the X direction, *G_y_*(*x*, *y*) is the gradient at point (x, y) in the Y direction, *G*_45°_(*x*, *y*) is the gradient at point (x, y) in the 45° direction, and *G*_135°_(*x*, *y*) is the gradient at point (x, y) in the 135° direction.

Finally, the gray value obtained by median filter with the 9 × 9 window was used to replace the above marked gray value. 

### 3.2. Comparison with Other Methods

Five advanced algorithms previously proposed by other scholars, as implemented in MATLAB, were considered for comparison in terms of calculating the average gray value and SI: Probability-based nonlocal means filtering [33], numerical multilook and 3D block-matching filtering [34], adaptive wavelet threshold processing [35], adaptive anisotropic diffusion [36], and Lee filtering. The average gray values (AGL) were processed to two decimal places and the SI was processed to four decimal places. The results reflect the measurement accuracy well, as shown in Table 1. The gray value of an image with a speckle variance of 0.9 was restored, as shown in Figure 6. In order to achieve a better and more intuitive response to the effect of the six algorithms, this paper intercepts the same parts of the gray value images recovered by the six algorithms and the grey value curve is analyzed in Figure 7. The red curve represents the gray value of the original image and the blue one represents the recovery value curve recovered by the five algorithms. 

Based on the statistical data from the experimental images, the curves showing the variation in the average gray value with the speckle intensity after the application of the six algorithms were obtained, as shown in Figure 8.

As shown in Figure 6, Figure 8, and Table 1, the algorithm proposed in this paper restored an image contaminated with speckle noise, achieving an SI value close to zero, indicating that the speckle noise was suppressed. Compared with the five previously proposed algorithms, the algorithm proposed in this paper showed a remarkable advantage in terms of recovery ability. To further verify the universality of the proposed algorithm, a restoration test was carried out on 17,500 images containing speckle noise in different formats and with different gray values and the recovery accuracy was 99.97%. Thus, it is shown that speckle noise can be effectively removed from images in different formats and with different gray values. 

## 4. Experimental Study

### 4.1. Application of PNLM-G to Refractive Index Change Detection of Reflected Light Images in PSi Microcavity Surface

In our previous study, when a laser light interacted with the surface of a PSi microcavity at 0° (vertical incidence), the reflection spectrum of the PSi microcavity obtained by increasing the incidence angle was the same as that obtained by increasing the refractive index [12]. Therefore, a change in the refractive index of a PSi microcavity can be considered equivalent to a change in the laser incidence angle. For a small angle of incidence, the enhancement of the reflected light intensity on the microcavity surface is proportional to the increase in the incidence angle [15], that is, the gray value of a digital image of the microcavity surface increases with an increasing incidence angle [15]. Here, we applied the proposed algorithm to detect refractive index changes in PSi microcavity images. Each microcavity unit is a circular unit region in the PSi microarray. 

The porous silicon microcavities were prepared by the anode electrochemical etching method. The silicon substrate was a P-type crystalline silicon (resistivity of 0.03–0.06 Ω⋅cm, crystal orientation of <100>). The electrolytic etching solution consisted of hydrogen fluoride acid (concentration of 40%) and anhydrous ethanol (C2H5OH, concentration ≥99%) with a volume ratio of 1:1. The sample electrochemical etching was controlled by Labview software and corroded at room temperature in a dark environment. The microcavity consisted of a defect layer sandwiched between two Bragg layers, an upper layer and a lower layer. The Bragg structures on both sides each contained 6 cycles. A current density of 110 mA/cm2 and an etching duration of 1.0 s were applied to obtain a low refractive index layer (*n*_L_ = 1.13) with a thickness of 140 nm. A current density of 60 mA/cm2 and an etching duration of 1.2 s were applied to obtain a high refractive index layer (*n*_H_ = 1.58) with a thickness of 100 nm. The current density and the etching duration were set to 110 mA/cm2 and 2 s to obtain a defect layer with a thickness of 560 nm. A pause of 3 s following each layer corrosion occurred in order to ensure relative uniform corrosion for each layer. A total of 25 dielectric layers was etched and the defect wavelength of PSM was 633 nm. 

The structure and measured reflection spectra of a PSi microcavity are illustrated in Figure 9 and Figure 10, respectively.

The gray value detection system based on the PSM surface image is shown in Figure 11. The light source was a He–Ne laser (633 nm, 1.8 mW). A1 and A2 were the apertures and lenses L1 and L2 were produced from a collimation beam expander system. The expanded beam passed through a beam splitter (5:5) onto a PSM microarray sample placed at the center of the goniometer. The sample could be rotated with the goniometer and the incident light was reflected to the digital microscope. CCD (1/3", 5 million pixels, CMOS, SONY) received light reflected from the surface of the PSi microcavity/PSi microcavity arrays. The optical lens magnification in the digital microscope was 0.7×–4.5× and was continuously adjustable.

In the detection device, a laser light with a wavelength of 633 nm interacted with the PSi microcavities at different angles and images of the device surface were obtained with a digital imaging device, as shown in Figure 12. 

First, the RGB image of the light reflected by the PSi microcavity at an incidence angle of 5° in Figure 12 was transformed into a gray value image and seven methods were used to restore the gray value, as shown in Figure 13. The proposed algorithm is effective for eliminating speckles that obscure the real image, can suppress the influence of speckle noise, and can restore the gray value of the image. We compared the results of our proposed algorithm to the results of five other advanced algorithms, including probability-based nonlocal mean filtering [33], numerical multilook and 3D block matching filtering [34], adaptive wavelet threshold processing [35], adaptive anisotropic diffusion [36], and Lee filtering. We found that the adaptive median filtering algorithm did not achieve the desired effect in removing real speckles. Additionally, it did not restore the true gray value of the original image. 

Based on the theoretical analysis presented in this paper and the observed effect of real speckle removal, it can be concluded that the proposed algorithm can effectively achieve gray value restoration. Therefore, we applied this algorithm to the images in Figure 12. The correspondence between the different incidence angles and gray values obtained after processing with the proposed algorithm is shown in Figure 14. 

The fitting formula and judgment coefficient before processing are, respectively, 

(17)$$Y=0.98X+0.40\;\mathrm{and}\;R^{2}=0.958.$$

The fitting formula and judgment coefficient after processing are, respectively,

(18)$$Y=1.01X+0.01\;\mathrm{and}\;R^{2}=0.9986.$$

It can be seen from Figure 14 that the linear fit to the data processed with our denoising algorithm was better than the fit to the unprocessed data. In addition, for the same change in gray value, the change in angle as evaluated with our denoising algorithm was larger than that achieved without processing. 

From the refractive indexes calculated based on the relationship between the changes in incidence angle and refractive index, the relationship between the changes in the gray value and the refractive index was obtained, as shown in Figure 15. There was a good linear relationship between the change in the refractive index and the change in the gray value. As seen from Figure 14 and Figure 15, after an image was processed with our denoising algorithm, the obtained change in the refractive index was larger than that obtained before processing for the same change in the gray value, demonstrating that the sensitivity of the refractive index detection was improved. 

It can be seen from the above experiment that the proposed algorithm is very suitable for recovering the gray values of images affected by speckle noise. The proposed algorithm can effectively reduce the influence of speckle noise in PSi microarray images and improve the detection accuracy achieved based on those images. 

### 4.2. Gray-Level Restoration of PSi Microcavity Array Images

A PSi microarray was fabricated on monocrystalline silicon by means of lithography and electrochemical etching technology [15]. Each unit in the array was a microcavity structure and the device parameters were the same as the microcavity parameters. Silicon nitride films with a thickness of 1.5 µm were deposited on silicon substrates by plasma-enhanced chemical vapor deposition (PECVD). Silicon nitride as a masking material has good corrosion resistance in the process of electrochemical etching. In the process of preparing porous silicon, the single crystal silicon below could not be etched within about 4 min. In order to enhance the adhesion between silicon nitride layer and photoresist, a layer of hexamethyldisiloxane (HMDS) was first coated on the silicon nitride film, then a layer of 1.1 m thick photoresist was coated on the HMDS. A mask (9 × 9 array, 300 µm in diameter and 200 µm in spacing) with the surface pattern of the array was used for exposure. Reactive ion etcher was used. After lithography, PSi microcavity arrays were obtained by electrochemical etching under the same conditions as those in Section 4.1. The structure of the PSi microcavity array is shown in Figure 16. 

Each array unit had a defect-state wavelength of 633 nm. Laser light with a wavelength of 633 nm interacted with the surface of the PSi microarray at different angles and images of the array surfaces were obtained by a digital microscope. Figure 17a shows a microarray image acquired with an incidence angle of 5°. The theoretical analysis presented above demonstrated that the proposed algorithm has the best ability to restore the gray values of such images. Therefore, the proposed algorithm was applied to an actual PSi microarray image to verify these results. In this experiment, we used previously described methods of preprocessing, tilt-correction, and sample segmentation [16]. Finally, we obtained the image of each PSi microcavity array unit, as shown in Figure 17b. 

By measuring the average gray value of each PSi microcavity array unit in Figure 17b, we obtained the overall average gray value and the SI of all array units. The measurement results are shown in Table 2 and Table 3. Table 2 shows the measurement results before filtering and Table 3 shows the average gray value of each PSi microcavity array unit after the application of the proposed algorithm. 

It can be seen from the above experiment that, after filtering with the proposed algorithm, the SI is close to zero, indicating that the gray value can be effectively recovered. The proposed algorithm has a better recovery ability than the other advanced algorithms considered. The proposed algorithm can effectively eliminate the influence of speckle noise and improve the detection accuracy of refractive index change in each PSi microcavity array unit. 

## 5. Conclusions

In order to study the influence of speckle noise on the gray value of the image, speckle noise with intensities of 0 to 1 was added to the gray value image and each group of gray values was experimented with 2500 images, respectively. Through drawing observations and data research, it was found that the image gray value increased with the increase of speckle noise intensity. Then, by further analyzing the gray value data of speckle noise image, it was found that the singular pixel points were the main factor that affected the change in gray value. Based on the above experiments, a new algorithm named PNLM-G is proposed to solve the influence of speckle noise on image gray value. PNLM-G algorithm adds the gradient operator on the basis of the PNLM algorithm. According to the gradient of the gray value of the image, the gray value of the remaining singular pixel points was marked. The gray value of the marked singular pixel points was taken as the center of the window of the median filter and the new central gray value of the median filter was used to replace the original central gray value to remove the influence of the singular pixel points on the gray value of the image. This made up for the shortcoming that the PNLM algorithm could not completely remove the influence of residual singular pixel points on the image gray value. Finally, by comparing this algorithm with other five algorithms, it was proved that the proposed algorithm is advanced and effective in suppressing speckle noise and restoring image gray value. PSi microcavity arrays were prepared and images of the light reflected from the arrays were obtained. The proposed algorithm was applied to the speckle denoising of images of the PSi microcavity and the PSi microcavity arrays, which corrected the gray value of the image and improved the accuracy of the refractive index change detection. At present, we are carrying out research on the detection of many different biological reactions on the same porous silicon PSM array by the gray value method.

## Figures and Tables

**Figure 1 sensors-19-02975-f001:**
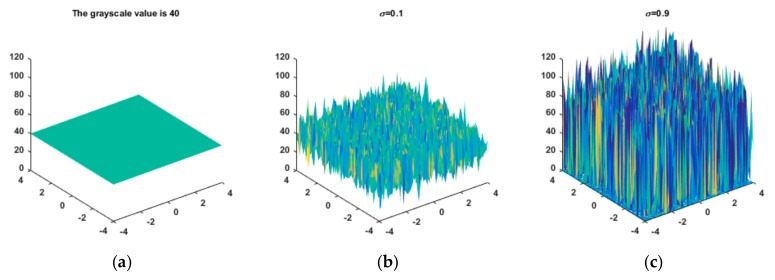
(**a**) The grayscale value is 40; (**b**) The effects of speckle noise with variances of 0.1 on image gray values; (**c**) The effects of speckle noise with variances of 0.9 on image gray values.

**Figure 2 sensors-19-02975-f002:**
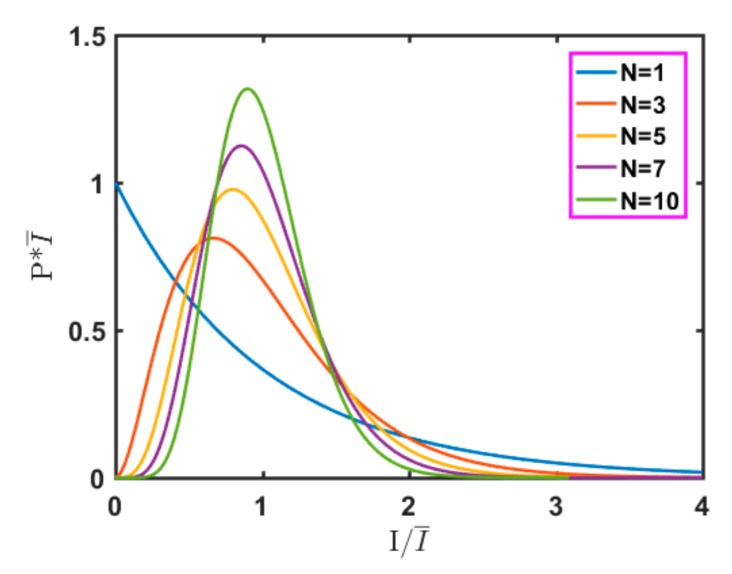
Probability density function of the sum of *N* independent equal-intensity speckle patterns.

**Figure 3 sensors-19-02975-f003:**
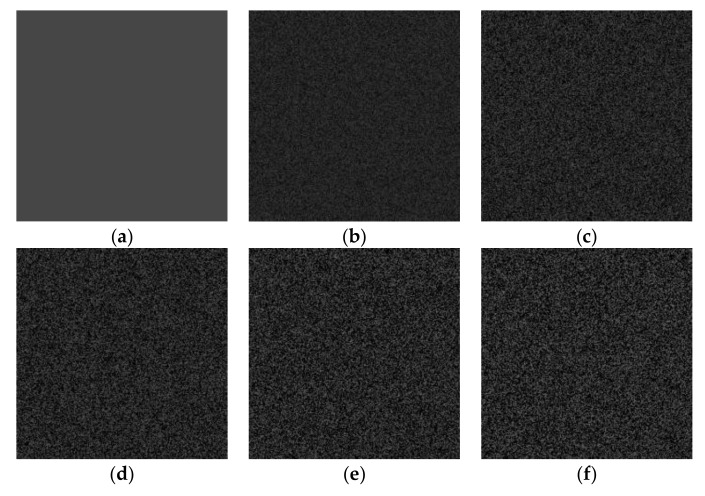
Experimental images generated by MATLAB: (**a**) Original image; (**b**) Var = 0.1; (**c**) Var = 0.3; (**d**) Var = 0.5; (**e**) Var = 0.7; (**f**) Var = 0.9.

**Figure 4 sensors-19-02975-f004:**
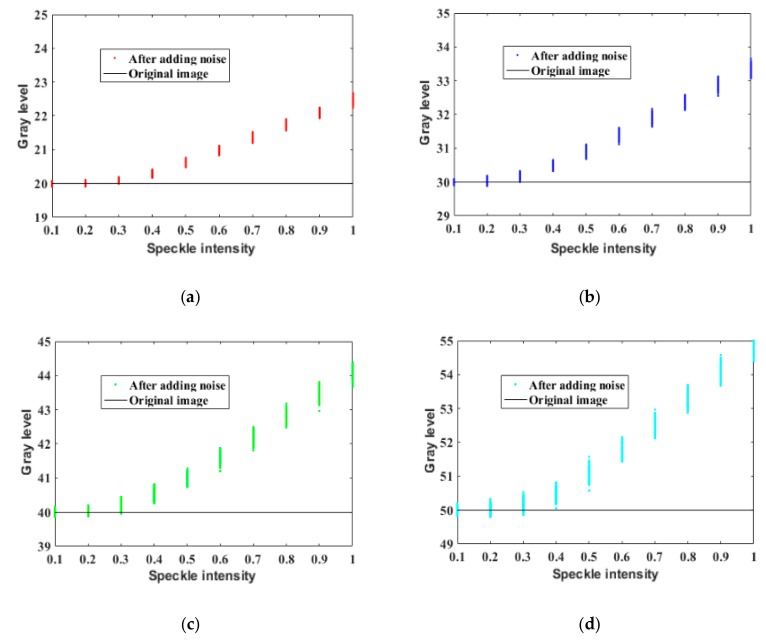
Influence of speckle noise on image gray values: (**a**) Gray value of 20; (**b**) gray value of 30; (**c**) gray value of 40; (**d**) gray value of 50.

**Figure 5 sensors-19-02975-f005:**
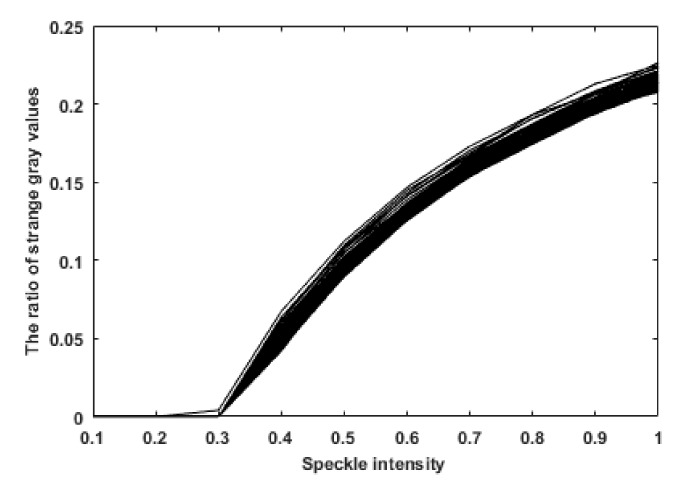
Relationship between the speckle intensity and the proportion of singular pixels.

**Figure 6 sensors-19-02975-f006:**
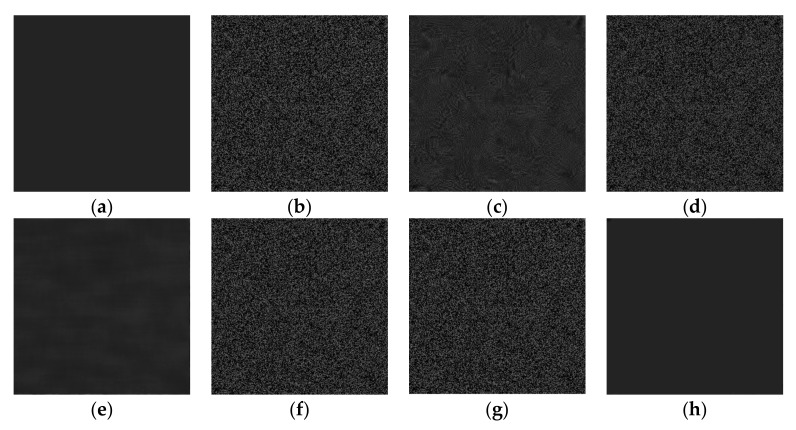
Six algorithm recovery effects: (**a**) Original image; (**b**) after adding noise; (**c**) BM3D; (**d**) Lee; (**e**) PNLM; (**f**) NAWT; (**g**) PM; (**h**) PNLM-G.

**Figure 7 sensors-19-02975-f007:**
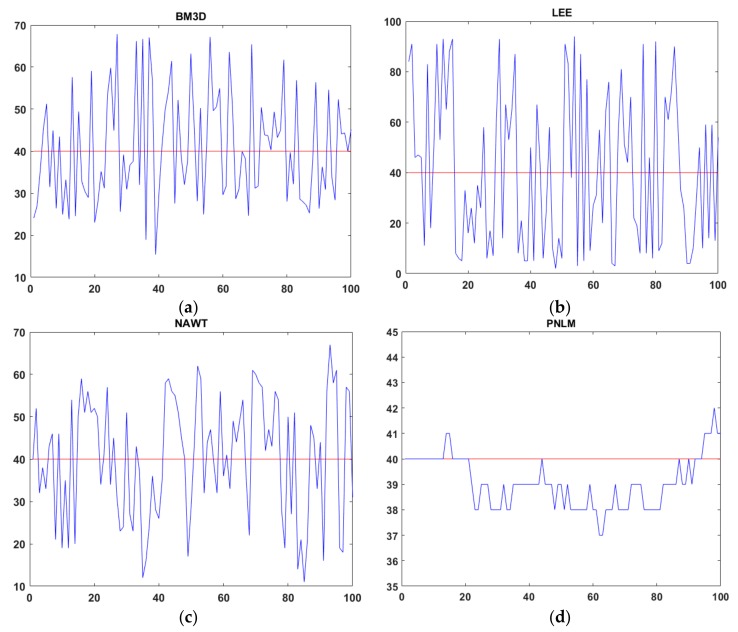
(**a**) Gray recovery of BM3D; (**b**) gray recovery of LEE; (**c**) gray recovery of NAWT; (**d**) gray recovery of PNLM; (**e**) gray recovery of PM; (**f**) gray recovery of PNLM-G.

**Figure 8 sensors-19-02975-f008:**
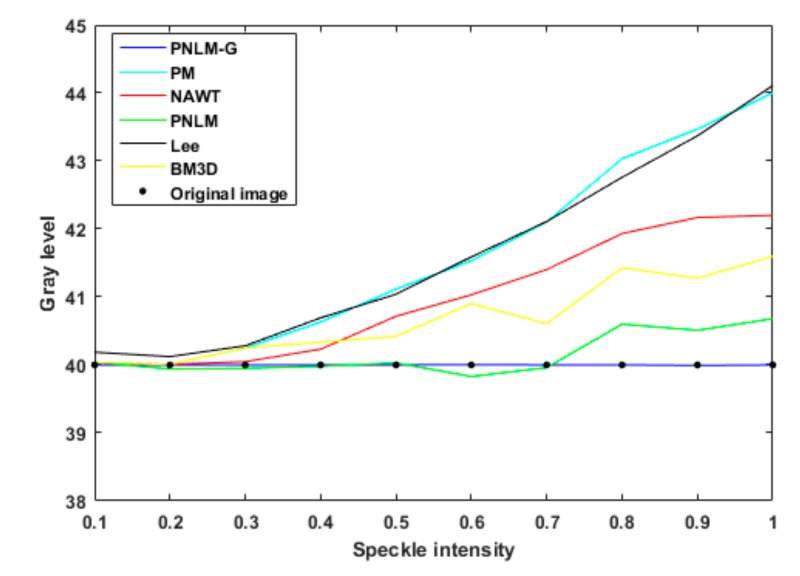
Comparison of the ability of the six algorithms to restore the gray value of an image.

**Figure 9 sensors-19-02975-f009:**
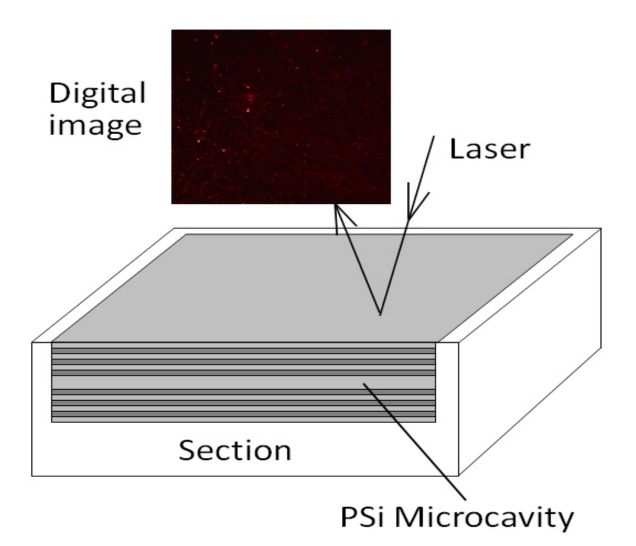
Digital image of light reflected from the surface of a PSi microcavity.

**Figure 10 sensors-19-02975-f010:**
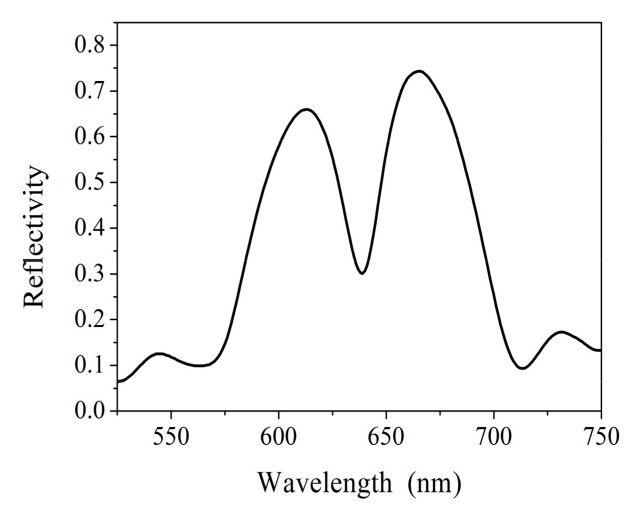
Reflection spectra of a measured porous silicon microcavity.

**Figure 11 sensors-19-02975-f011:**
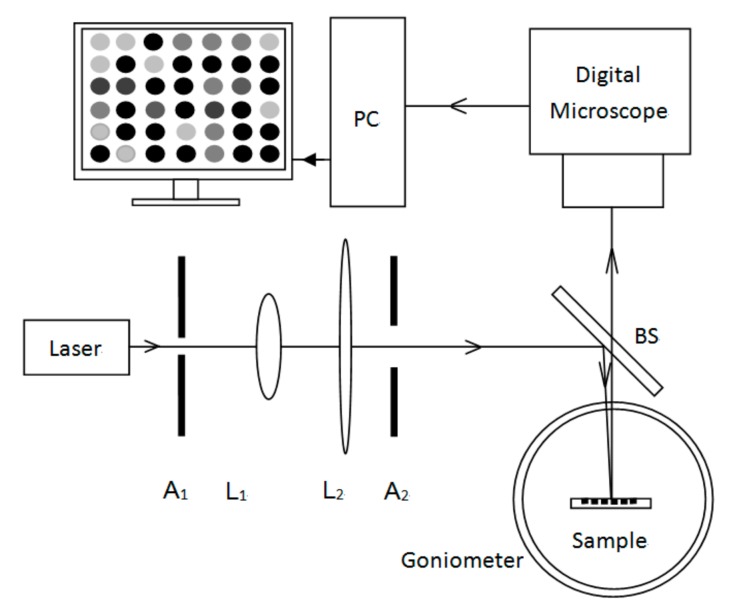
Gray value detection system based on PSM/PSM microarray surface image.

**Figure 12 sensors-19-02975-f012:**
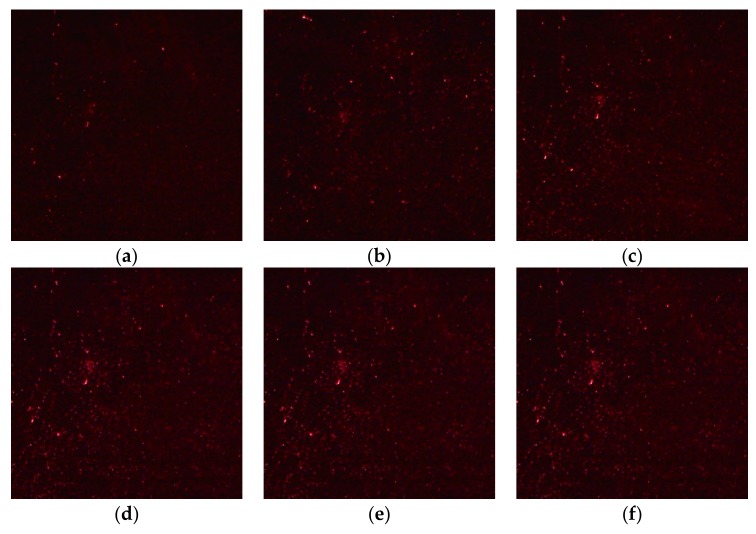
Reflected images of PSi microcavities at different incident angles: (**a**) 0°; (**b**) 1°; (**c**) 2°; (**d**) 3°; (**e**) 4°; (**f**) 5°.

**Figure 13 sensors-19-02975-f013:**
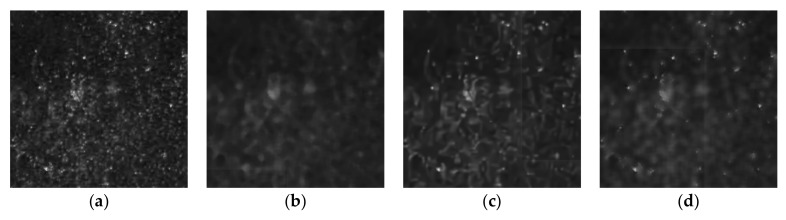
The restoration of gray value of PSi microcavity with an incident angle of 5°. (**a**) Original image; (**b**) adaptive median filter; (**c**) BM3D; (**d**) Lee; (**e**) PNLM; (**f**) NAWT; (**g**) PM; (**h**) PNLM-G.

**Figure 14 sensors-19-02975-f014:**
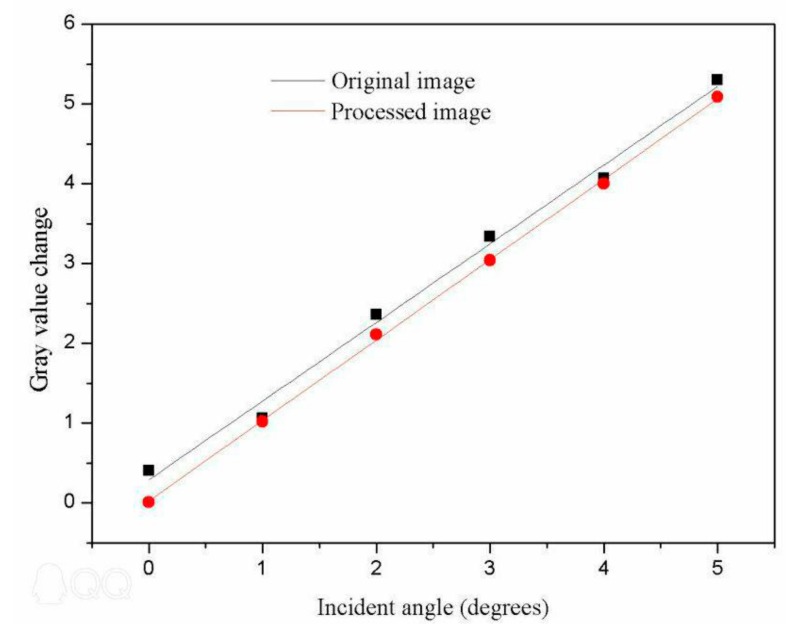
Relationship between the gray value of a PSi microcavity surface image and the laser incidence angle.

**Figure 15 sensors-19-02975-f015:**
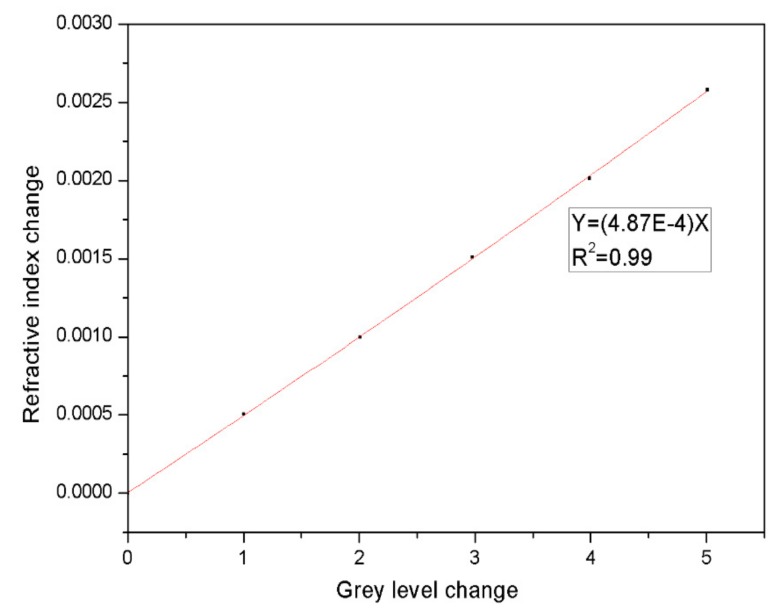
Relationship between the change in refractive index and the change in grey level in the PSi microcavity.

**Figure 16 sensors-19-02975-f016:**
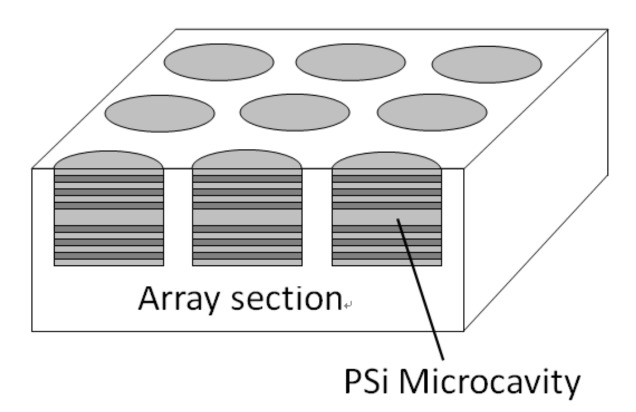
The structure of the PSi microcavity arrays.

**Figure 17 sensors-19-02975-f017:**
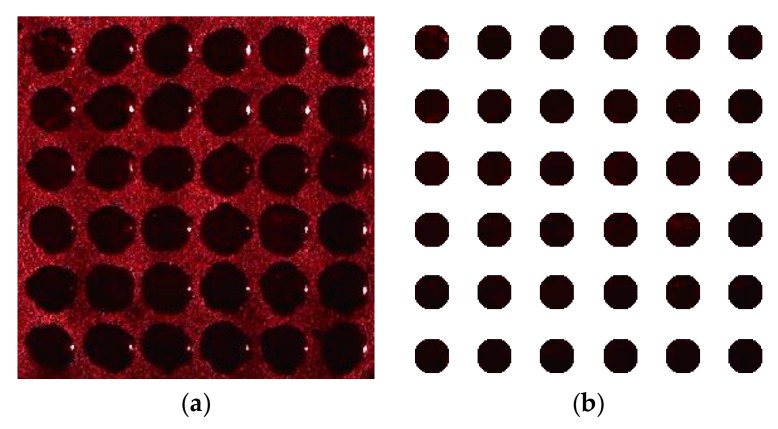
(**a**) Actual PSi microarray image obtained with digital imaging equipment; (**b**) measured PSi microarray image after pretreatment, correction, and segmentation.

**Table 1 sensors-19-02975-t001:** Data statistics of ten pictures.

Speckle Intensity	Index	UnprocesSed Image	Lee	BM3D	NAWT	PNLM	PM	PNLM-G
0.10	AGL	40.03	40.19	40.03	40.02	40.04	40.03	40.00
	SI^	0.3236	0.0470	0.0263	0.0593	0.0364	0.3233	0.0017
0.20	AGL	40.01	40.12	40.00	40.01	39.94	40.00	40.00
	SI	0.4792	0.2520	0.0354	0.0658	0.0234	0.4800	0.0033
0.30	AGL	40.25	40.28	40.25	40.05	39.95	40.24	40.00
	SI	0.5784	0.3880	0.0443	0.0789	0.0297	0.5778	0.0021
0.40	AGL	40.63	40.69	40.34	40.23	39.98	40.63	40.00
	SI	0.6431	0.4708	0.0572	0.0846	0.0340	0.6431	0.0026
0.50	AGL	41.11	41.04	40.42	40.72	40.03	41.12	40.00
	SI	0.6905	0.5300	0.0795	0.0903	0.0349	0.6885	0.0020
0.60	AGL	41.53	41.59	40.90	41.03	39.83	41.53	40.00
	SI	0.7249	0.5718	0.1127	0.0963	0.0424	0.7276	0.0051
0.70	AGL	42.10	42.11	40.61	41.40	39.96	42.10	40.00
	SI	0.7560	0.6089	0.1522	0.1012	0.0515	0.7545	0.0041
0.80	AGL	43.03	42.76	41.43	41.93	40.60	43.03	40.00
	SI	0.7779	0.6348	0.1931	0.1027	0.0444	0.7734	0.0051
0.90	AGL	43.47	43.37	41.27	42.17	40.51	43.47	39.99
	SI	0.7960	0.6560	0.2321	0.1048	0.0484	0.7957	0.0090
1.00	AGL	44.00	44.11	41.60	42.20	40.68	44.00	40.00
	SI	0.8120	0.6746	0.2741	0.1068	0.0434	0.8146	0.0121

**Table 2 sensors-19-02975-t002:** The average gray value corresponding to the sample point before filtering.

	Column	First Column	Second Column	Third Column	Fourth Column	Fifth Column	Sixth Column
Row							
First row		17.74	16.00	15.98	16.20	16.03	15.95
Second row		16.28	15.99	16.01	16.06	16.13	15.98
Third row		15.93	16.12	15.99	15.98	15.99	16.25
Fourth row		15.83	15.94	16.12	15.98	16.37	15.99
Fifth row		15.97	15.99	15.95	15.99	16.16	16.03
Sixth row		15.94	15.99	16.00	15.93	15.97	15.93
Average grey level		16.07					
Speckle index		0.5069					

**Table 3 sensors-19-02975-t003:** The average gray value corresponding to the sample point after filtering.

	Column	First Column	Second Column	Third Column	Fourth Column	Fifth Column	Sixth Column
Row							
First row		16.01	16.00	16.00	16.00	16.00	16.00
Second row		16.00	16.00	16.00	16.00	16.00	16.00
Third row		16.00	16.00	16.00	16.00	16.00	16.00
Fourth row		16.00	16.00	16.00	16.00	16.01	16.00
Fifth row		16.00	16.00	16.00	16.00	16.00	16.00
Sixth row		15.99	16.00	16.00	15.99	16.00	16.00
Average grey level		16.00					
Speckle index		0.0046					

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
