# Peer review of "Applying Speckle Noise Suppression to Refractive Indices Change Detection in Porous Silicon Microarrays"

_sensors, 2019, doi:10.3390/s19132975_

Round 1

Reviewer 1 Report

Authors present a paper that must be improved before being considered for submission. First of all, in the paper there is a lot of stuff that could be suitable for a journal more specialized in algorithms than in sensors. On the contrary, there is not any really sensing application that could be of interest for the readers. Porous silicon micro cavities make sense if they are read in white light reflection, otherwise a single layer of porous silicon could be even better. Authors do not report any detail on fabrication and on the characteristic of the porous silicon microcavity: which is the resonance peak? It is important to show the full reflection spectrum since from this it can be predicted the laser reflection. The optical characterization should be completely changed and improved. For example, Fig. 11 is only a demonstration of scattering increasing and not refractive index increasing. The work done by the authors could have sense if applied to multiwavelength images of porous silicon microarray and in presence of a volatile substance (like ethanol) which could prove the sensing ability of their technique. In the present form, the paper should be rejected.

Author Response

Dear Reviewer 

ReApplying speckle noise suppression to refractive indices change detection in porous silicon microarrays( (Manuscript ID: sensors-517939)

Thank you very much for your letter and the comments from the referees about our paper submitted to Remote Sensors.

We have learned much from the reviewers’ comments, which are fair, encouraging and constructive.  

After carefully studying the reviewer' comments and your advice, we have made corresponding changes to the paper. I hope this will make it more acceptable for publication.

Our response of the comments is enclosed at the end of this letter.

If you have any question about this paper, please don’t hesitate to contact us via email: jzhh@xju.edu.cn, Tel.: +8613899985599, and Fax: 0991-8583362.

Yours sincerely

  Ruyong Ren

Reviewer 2 Report

The respected authors present a method to reduce speckle noise for porous silicon microarray biosensors. The presented algorithm allows to reliably recover the gray value of the PSi microarray image. The method has been compared to other methods and has been verified with real data and actual PSi microarray images.

The manuscript is well structured, in a clear and comprehensive introduction into the field (Maybe an image or sketch (similar to the one in Figure 15) would probably be helpful for novices in the field.), followed by a theoretical and experimental analysis. While the theoretical analysis could use some more details (or added information on the algorithm potentially in supplementary material or even in providing the algorithm itself), the experimental section should contain more information on Materials & Methods. How exactly where the PSi microarrays fabricated (or reference), how was the images acquired (sensor type, lenses,…), how where the measurement angles acquired,…

Summary and conclusion is a little too short. Outlook missing. Please revisit.

Some more specific requests:

-          L35/36: “… first proposed in the 1990.” Please add reference.

-          L37/38: not all fluorescent markers are expensive.

-          L52/53 and 55: 10-4 is actually not very high. Surface Plasmon resonance, integrated optical interferometers/resonators and waveguide grating couplers have refractive index sensitivities in the order of 10-7 to 10-9. Wording should therefore he changed or put in perspective.

-          L76: do you mean “accuracy” or precision? Or both?

-          L132ff: add reference(s).

-          Figure 5: unclear what the individual lines are.

-          L244/245: why does it have to be between [0.01. 0.02]? Unclear. Please describe.

-          L245: “convey the wrong message”. What “wrong message”? Consider rephrasing.

-          L246: “it is very difficult”. What is very difficult? Please specify.

-          L288ff. Consider rephrasing. “Figure 7” should be mentioned before or in the corresponding sentence to avoid confusion.

-          Figure 7: I’d personally use the same scale for all images to illustrate the effect better and make it easier to visually compare the methods, but it’s up to the authors.

-          Figure 9 is not useful, unless one zooms more into on the y-axis.

-          Was the refractive index measured directly, only derived from angle measurements? What’s the limit of detection?

-          There seems to be a bias in the selection of references. There’s been a tremendous amount of work in western literature on PSi microarrays and label-free optical biosensing in general (even has its origins there), which seems mostly disregarded. Please add the corresponding references.

-          Affiliations in name 1, 3 and 4, but listed Institutes only 1, 2, 3.

-          Text needs some editing, often “space” missing between numbered items such as titles or figures.

Thanks.

Author Response

(The authors gave the same response as above.)

Round 2

Reviewer 1 Report

In the revised form the paper can be accepted for publication